# Differential Privacy without Sensitivity

**Kentaro Minami**
The University of Tokyo
kentaro_minami@mist.i.u-tokyo.ac.jp

**Hiromi Arai**
The University of Tokyo
arai@dl.itc.u-tokyo.ac.jp

**Issei Sato**
The University of Tokyo
sato@k.u-tokyo.ac.jp

**Hiroshi Nakagawa**
The University of Tokyo
nakagawa@dl.itc.u-tokyo.ac.jp

## Abstract

The exponential mechanism is a general method to construct a randomized estimator that satisfies $(\varepsilon, 0)$-differential privacy. Recently, Wang et al. showed that the Gibbs posterior, which is a data-dependent probability distribution that contains the Bayesian posterior, is essentially equivalent to the exponential mechanism under certain boundedness conditions on the loss function. While the exponential mechanism provides a way to build an $(\varepsilon, 0)$-differential private algorithm, it requires boundedness of the loss function, which is quite stringent for some learning problems. In this paper, we focus on $(\varepsilon, \delta)$-differential privacy of Gibbs posteriors with convex and Lipschitz loss functions. Our result extends the classical exponential mechanism, allowing the loss functions to have an unbounded sensitivity.

## 1 Introduction

Differential privacy is a notion of privacy that provides a statistical measure of privacy protection for randomized statistics. In the field of privacy-preserving learning, constructing estimators that satisfy $(\varepsilon, \delta)$-differential privacy is a fundamental problem. In recent years, differentially private algorithms for various statistical learning problems have been developed [8, 14, 3].

Usually, the estimator construction procedure in statistical learning contains the following minimization problem of a data-dependent function. Given a dataset $D_n = \{x_1, \ldots, x_n\}$, a statistician chooses a parameter $\theta$ that minimizes a cost function $\mathcal{L}(\theta, D_n)$. A typical example of cost function is the empirical risk function, that is, a sum of *loss function* $\ell(\theta, x_i)$ evaluated at each sample point $x_i \in D_n$. For example, the maximum likelihood estimator (MLE) is given by the minimizer of empirical risk with loss function $\ell(\theta, x) = -\log p(x \mid \theta)$.

To achieve a differentially private estimator, one natural idea is to construct an algorithm based on a *posterior sampling*, namely drawing a sample from a certain data-dependent probability distribution. The exponential mechanism [16], which can be regarded as a posterior sampling, provides a general method to construct a randomized estimator that satisfies $(\varepsilon, 0)$-differential privacy. The probability density of the output of the exponential mechanism is proportional to $\exp(-\beta\mathcal{L}(\theta, D_n))\pi(\theta)$, where $\pi(\theta)$ is an arbitrary prior density function, and $\beta > 0$ is a parameter that controls the degree of concentration. The resulting distribution is highly concentrated around the minimizer $\theta^* \in \operatorname{argmin}_\theta \mathcal{L}(\theta, D_n)$. Note that most differential private algorithms involve a procedure to add some noise (e.g. the Laplace mechanism [12], objective perturbation [8, 14], and gradient perturbation [3]), while the posterior sampling explicitly designs the density of the output distribution.

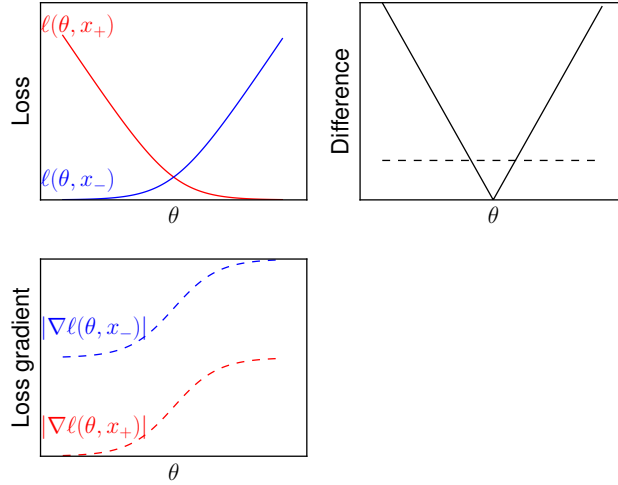

Figure 1: An example of a logistic loss function $\ell(\theta, x) := \log(1 + \exp(-y\theta^\top z))$. Considering two points $x_\pm = (z, \pm 1)$, the difference of the loss $|\ell(\theta, x_+) - \ell(\theta, x_-)|$ increases proportionally to the size of the parameter space (solid lines). In this case, the value of the $\beta$ in the exponential mechanism, which is inversely proportional to the maximum difference of the loss function, becomes very small. On the other hand, the difference of the gradient $|\nabla\ell(\theta, x_+) - \nabla\ell(\theta, x_-)|$ does not exceed twice of the Lipschitz constant (dashed lines). Hence, our analysis based on Lipschitz property does not be influenced by the size of the parameter space.

Table 1: Regularity conditions for $(\varepsilon, \delta)$-differential privacy of the Gibbs posterior. Instead of the boundedness of the loss function, our analysis in Theorem 7 requires its Lipschitz property and convexity. Unlike the classical exponential mechanism, our result explains "shrinkage effect" or "contraction effect", namely, the upper bound for $\beta$ depends on the concavity of the prior $\pi$ and the size of the dataset $n$.

|  | $(\varepsilon, \delta)$ | Loss function $\ell$ | Prior $\pi$ | Shrinkage |
|---|---|---|---|---|
| Exponential mechanism [16] | $\delta = 0$ | Bounded sensitivity | Arbitrary | No |
| Theorem 7 | $\delta > 0$ | Lipschitz and convex | Log-concave | Yes |
| Theorem 10 | $\delta > 0$ | Bounded, Lipschitz and strongly convex | Log-concave | Yes |

We define the density of the *Gibbs posterior distribution* as

$$G_\beta(\theta \mid D_n) := \frac{\exp(-\beta \sum_{i=1}^n \ell(\theta, x_i))\pi(\theta)}{\int \exp(-\beta \sum_{i=1}^n \ell(\theta, x_i))\pi(\theta)\mathrm{d}\theta}. \tag{1}$$

The Gibbs posterior plays important roles in several learning problems, especially in PAC-Bayesian learning theory [6, 21]. In the context of differential privacy, Wang et al. [20] recently pointed out that the Bayesian posterior, which is a special version of (1) with $\beta = 1$ and a specific loss function, satisfies $(\varepsilon, 0)$-differential privacy because it is equivalent to the exponential mechanism under a certain regularity condition. Bassily, et al. [3] studied an application of the exponential mechanism to private convex optimization.

In this paper, we study the $(\varepsilon, \delta)$-differential privacy of the posterior sampling with $\delta > 0$. In particular, we consider the following statement.

**Claim 1.** Under a suitable condition on loss function $\ell$ and prior $\pi$, there exists an upper bound $B(\varepsilon, \delta) > 0$, and the Gibbs posterior $G_\beta(\theta \mid D_n)$ with $\beta \leq B(\varepsilon, \delta)$ satisfies $(\varepsilon, \delta)$-differential privacy. The value of $B(\varepsilon, \delta)$ does not depend on the boundedness of the loss function.

We point out here the analyses of $(\varepsilon, 0)$-differential privacy and $(\varepsilon, \delta)$-differential privacy with $\delta > 0$ are conceptually different in the regularity conditions they require. On one hand, the exponential mechanism essentially requires the boundedness of the loss function to satisfy $(\varepsilon, 0)$-differential privacy. On the other hand, the boundedness is not a necessary condition in $(\varepsilon, \delta)$-differential privacy. In this paper, we give a new sufficient condition for $(\varepsilon, \delta)$-differential privacy based on the convexity and the Lipschitz property. Our analysis widens the application ranges of the exponential mechanism in the following aspects (See also Table 1).

- (Removal of boundedness assumption) If the loss function is unbounded, which is usually the case when the parameter space is unbounded, the Gibbs posterior does not satisfy $(\varepsilon, 0)$-differential privacy in general. Still, in some cases, we can build an $(\varepsilon, \delta)$-differential private estimator.

- (Tighter evaluation of $\beta$) Even when the difference of the loss function is bounded, our analysis can yield a better scheme in determining the appropriate value of $\beta$ for a given privacy level. Figure 1 shows an example of logistic loss.

- (Shrinkage and contraction effect) Intuitively speaking, the Gibbs posterior becomes robust against a small change of the dataset, if the prior $\pi$ has a strong shrinkage effect (e.g. a Gaussian prior with a small variance), or if the size of the dataset $n$ tends to infinity. In our analysis, the upper bound of $\beta$ depends on $\pi$ and $n$, which explains such shrinkage and contraction effects.

## 1.1 Related work

$(\varepsilon, \delta)$-differential privacy of Gibbs posteriors has been studied by several authors. Mir ([18], Chapter 5) proved that a Gaussian posterior in a specific problem satisfies $(\varepsilon, \delta)$-differential privacy. Dimitrakakis et al. [10] considered Lipschitz-type sufficient conditions, yet their result requires some modification of the definition of the neighborhood on the database.

In general, the utility of sensitivity-based methods suffers from the size of the parameter space $\Theta$. Thus, getting around the dependency on the size of $\Theta$ is a fundamental problem in the study of differential privacy. For discrete parameter spaces, a general range-independent algorithm for $(\varepsilon, \delta)$-differential private maximization was developed in [7].

## 1.2 Notations

The set of all probability measures on a measurable space $(\Theta, \mathcal{T})$ is denoted by $\mathcal{M}_+^1(\Theta)$. A map between two metric spaces $f : (X, d_X) \to (Y, d_Y)$ is said to be $L$-Lipschitz, if $d_Y(f(x_1), f(x_2)) \leq L d_X(x_1, x_2)$ holds for all $x_1, x_2 \in X$. Let $f$ be a twice continuously differentiable function $f$ defined on a subset of $\mathbb{R}^d$. $f$ is said to be $m(> 0)$-strongly convex, if the eigenvalues of its Hessian $\nabla^2 f$ are bounded by $m$ from below. $f$ is said to be $M$-smooth,

# 2 Differential privacy with sensitivity

In this section, we review the definition of $(\varepsilon, \delta)$-differential privacy and the exponential mechanism.

## 2.1 Differential privacy

Differential privacy is a notion of privacy that provides a degree of privacy protection in a statistical sense. More precisely, differential privacy defines a closeness between any two output distributions that correspond to adjacent datasets.

In this paper, we assume that a dataset $D = D_n = (x_1, \dots, x_n)$ is a vector that consists of $n$ points in abstract attribute space $\mathcal{X}$, where each entry $x_i \in \mathcal{X}$ represents information contributed by one individual. Two datasets $D, D'$ are said to be adjacent if $d_H(D, D') = 1$, where $d_H$ is the Hamming distance defined on the space of all possible datasets $\mathcal{X}^d$.

We describe the definition of differential privacy in terms of *randomized estimators*. A randomized estimator is a map $\rho : \mathcal{X}^n \to \mathcal{M}_+^1(\Theta)$ from the space of datasets to the space of probability measures.

**Definition 2** (Differential privacy). Let $\varepsilon > 0$ and $\delta \geq 0$ be given privacy parameters. We say that a randomized estimator $\rho : \mathcal{X}^n \to \mathcal{M}_+^1(\Theta)$ satisfies $(\varepsilon, \delta)$-differential privacy, if for any adjacent datasets $D, D' \in \mathcal{X}^n$, an inequality

$$\rho_D(A) \leq e^\varepsilon \rho_{D'}(A) + \delta \tag{2}$$

holds for every measurable set $A \subset \Theta$.

## 2.2 The exponential mechanism

The exponential mechanism [16] is a general construction of $(\varepsilon, 0)$-differentially private distributions. For an arbitrary function $\mathcal{L} : \Theta \times \mathcal{X}^n \to \mathbb{R}$, we define the sensitivity by

$$\Delta_\mathcal{L} := \sup_{\substack{D, D' \in \mathcal{X}^n: \\ d_H(D, D') = 1}} \sup_{\theta \in \Theta} |\mathcal{L}(\theta, D) - \mathcal{L}(\theta, D')|, \tag{3}$$

which is the largest possible difference of two adjacent functions $f(\cdot, D)$ and $f(\cdot, D')$ with respect to supremum norm.

**Theorem 3** (McSherry and Talwar). Suppose that the sensitivity of the function $\mathcal{L}(\theta, D_n)$ is finite. Let $\pi$ be an arbitrary base measure on $\Theta$. Take a positive number $\beta$ so that $\beta \leq \varepsilon/2\Delta_\mathcal{L}$. Then a probability distribution whose density with respect to $\pi$ is proportional to $\exp(-\beta\mathcal{L}(\theta, D_n))$ satisfies $(\varepsilon, 0)$-differential privacy.

We consider the particular case that the cost function is given as sum form $\mathcal{L}(\theta, D_n) = \sum_{i=1}^n \ell(\theta, x_i)$. Recently, Wang et al. [20] examined two typical cases in which $\Delta_\mathcal{L}$ is finite. The following statement slightly generalizes their result.

**Theorem 4** (Wang, et al.). (a) Suppose that the loss function $\ell$ is bounded by $A$, namely $|\ell(\theta, x)| \leq A$ holds for all $x \in \mathcal{X}$ and $\theta \in \Theta$. Then $\Delta_\mathcal{L} \leq 2A$, and the Gibbs posterior (1) satisfies $(4\beta A, 0)$-differential privacy.

(b) Suppose that for any fixed $\theta \in \Theta$, the difference $|\ell(\theta, x_1) - \ell(\theta, x_2)|$ is bounded by $L$ for all $x_1, x_2 \in \mathcal{X}$. Then $\Delta_\mathcal{L} \leq L$, and the Gibbs posterior (1) satisfies $(2\beta L, 0)$-differential privacy.

The condition $\Delta_\mathcal{L} < \infty$ is crucial for Theorem 3 and cannot be removed. However, in practice, statistical models of interest do not necessarily satisfy such boundedness conditions. Here we have two simple examples of Gaussian and Bernoulli mean estimation problems, in which the sensitivities are unbounded.

- (Bernoulli mean) Let $\ell(p, x) = -x \log p - (1 - x) \log(1 - p)$ ($p \in (0, 1)$, $x \in \{0, 1\}$) be the negative log-likelihood of the Bernoulli distribution. Then $|\ell(p, 0) - \ell(p, 1)|$ is unbounded.

- (Gaussian mean) Let $\ell(\theta, x) = \frac{1}{2}(\theta - x)^2$ ($\theta \in \mathbb{R}$, $x \in \mathbb{R}$) be the negative log-likelihood of the Gaussian distribution with a unit variance. Then $|\ell(\theta, x) - \ell(\theta, x')|$ is unbounded if $x \neq x'$.

Thus, in the next section, we will consider an alternative proof technique for $(\varepsilon, \delta)$-differential privacy so that it does not require such boundedness conditions.

## 3 Differential privacy without sensitivity

In this section, we state our main results for $(\varepsilon, \delta)$-differential privacy in the form of Claim 1.

There is a well-known sufficient condition for the $(\varepsilon, \delta)$-differential privacy:

**Theorem 5** (See for example Lemma 2 of [13]). Let $\varepsilon > 0$ and $\delta > 0$ be privacy parameters. Suppose that a randomized estimator $\rho : \mathcal{X}^n \to \mathcal{M}_+^1(\Theta)$ satisfies a tail-bound inequality of log-density ratio

$$\rho_D \left\{ \log \frac{\mathrm{d}\rho_D}{\mathrm{d}\rho_{D'}} \geq \varepsilon \right\} \leq \delta \tag{4}$$

for every adjacent pair of datasets $D, D'$. Then $\rho$ satisfies $(\varepsilon, \delta)$-differential privacy.

To control the tail behavior (4) of the log-density ratio function $\log \frac{\mathrm{d}\rho_D}{\mathrm{d}\rho_{D'}}$, we consider the concentration around its expectation. Roughly speaking, inequality (4) holds if there exists an increasing function $\alpha(t)$ that satisfies an inequality

$$\forall t > 0, \quad \rho_D \left\{ \log \frac{\mathrm{d}\rho_D}{\mathrm{d}\rho_{D'}} \geq D_{\mathrm{KL}}(\rho_D, \rho_{D'}) + t \right\} \leq \exp(-\alpha(t)), \tag{5}$$

where $\log \frac{\mathrm{d}G_{\beta,D}}{\mathrm{d}G_{\beta,D'}}$ is the log-density ratio function, and $D_{\mathrm{KL}}(\rho_D, \rho_{D'}) := \mathbb{E}_{\rho_D} \log \frac{\mathrm{d}\rho_D}{\mathrm{d}\rho_{D'}}$ is the Kullback-Leibler (KL) divergence. Suppose that the Gibbs posterior $G_{\beta,D}$, whose density $G(\theta \mid D)$ is defined by (1), satisfies an inequality (5) for a certain $\alpha(t) = \alpha(t, \beta)$. Then $G_{\beta,D}$ satisfies (4) if there exist $\beta, t > 0$ that satisfy the following two conditions.

1. KL-divergence bound: $D_{\mathrm{KL}}(G_{\beta,D}, G_{\beta,D'}) + t \leq \varepsilon$
2. Tail-probability bound: $\exp(-\alpha(t, \beta)) \leq \delta$

## 3.1 Convex and Lipschitz loss

Here, we examine the case in which the loss function $\ell$ is Lipschitz and convex, and the parameter space $\Theta$ is the entire Euclidean space $\mathbb{R}^d$. Due to the unboundedness of the domain, the sensitivity $\Delta_{\mathcal{L}}$ can be infinite, in which case the exponential mechanism cannot be applied.

**Assumption 6.** (i) $\Theta = \mathbb{R}^d$.

(ii) For any $x \in \mathcal{X}$, $\ell(\cdot, x)$ is non-negative, $L$-Lipschitz and convex.

(iii) $-\log \pi(\cdot)$ is twice differentiable and $m_\pi$-strongly convex.

In Assumption 6, the loss function $\ell(\cdot, x)$ and the difference $|\ell(\cdot, x_1) - \ell(\cdot, x_2)|$ can be unbounded. Thus, the classical argument of the exponential mechanism in Section 2.2 cannot be applied. Nevertheless, our analysis shows that the Gibbs posterior satisfies $(\varepsilon, \delta)$-differential privacy.

**Theorem 7.** Let $\beta \in (0, 1]$ be a fixed parameter, and $D, D' \in \mathcal{X}^n$ be an adjacent pair of datasets. Under Assumption 6, inequality

$$G_{\beta,D} \left\{ \log \frac{\mathrm{d}G_{\beta,D}}{\mathrm{d}G_{\beta,D'}} \geq \varepsilon \right\} \leq \exp \left( -\frac{m_\pi}{8L^2\beta^2} \left( \varepsilon - \frac{2L^2\beta^2}{m_\pi} \right)^2 \right) \tag{6}$$

holds for any $\varepsilon > \frac{2L^2\beta^2}{m_\pi}$.

Gibbs posterior $G_{\beta,D}$ satisfies $(\varepsilon, \delta)$-differential privacy if $\beta > 0$ is taken so that the right-hand side of (6) is bounded by $\delta$. It is elementary to check the following statement:

**Corollary 8.** Let $\varepsilon > 0$ and $0 < \delta < 1$ be privacy parameters. Taking $\beta$ so that it satisfies

$$\beta \leq \frac{\varepsilon}{2L} \sqrt{\frac{m_\pi}{1 + 2\log(1/\delta)}}, \tag{7}$$

Gibbs posterior $G_{\beta,D}$ satisfies $(\varepsilon, \delta)$-differential privacy.

Note that the right-hand side of (6) depends on the strong concavity $m_\pi$. The strong concavity parameter corresponds to the precision (i.e. inverse variance) of the Gaussian, and a distribution with large $m_\pi$ becomes spiky. Intuitively, if we use a prior that has a strong shrinkage effect, then the posterior becomes robust against a small change of the dataset, and consequently the differential privacy can be satisfied with a little effort. This observation is justified in the following sense: the upper bound of $\beta$ grows proportionally to $\sqrt{m_\pi}$. In contrast, the classical exponential mechanism does not have that kind of prior-dependency.

## 3.2 Strongly convex loss

Let $\tilde{\ell}$ be a strongly convex function defined on the entire Euclidean space $\mathbb{R}^d$. If $\ell$ is a restriction of $\tilde{\ell}$ to a compact $L_2$-ball, the Gibbs posterior can satisfy $(\varepsilon, 0)$-differential privacy with a certain privacy level $\varepsilon > 0$ because of the boundedness of $\ell$. However, using the boundedness of $\nabla \ell$ rather than that of $\ell$ itself, we can give another guarantee for $(\varepsilon, \delta)$-differential privacy.

**Assumption 9.** Suppose that a function $\tilde{\ell} : \mathbb{R}^d \times \mathcal{X} \to \mathbb{R}$ is a twice differentiable and $m_\ell$-strongly convex with respect to its first argument. Let $\tilde{\pi}$ be a finite measure over $\mathbb{R}^d$ that $-\log \tilde{\pi}(\cdot)$ is twice differentiable and $m_\pi$-strongly convex. Let $\tilde{G}_{\beta,D}$ is a Gibbs posterior on $\mathbb{R}^d$ whose density with respect to the Lebesgue measure is proportional to $\exp(-\beta \sum_i \tilde{\ell}(\theta, x_i)) \tilde{\pi}(\theta)$. Assume that the mean of $\tilde{G}_{\beta,D}$ is contained in a $L_2$-ball of radius $\kappa$:

$$\forall D \in \mathcal{X}^n, \quad \left\| \mathbb{E}_{\tilde{G}_{\beta,D}}[\theta] \right\|_2 \leq \kappa. \tag{8}$$

Define a positive number $\alpha > 1$. Assume that $(\Theta, \ell, \pi)$ satisfies the following conditions.

(i) $\Theta$ is a compact $L_2$-ball centered at the origin, and its radius $R_\Theta$ satisfies $R_\Theta \leq \kappa + \alpha \sqrt{d/m_\pi}$.

(ii) For any $x \in \mathcal{X}$, $\ell(\cdot, x)$ is $L$-Lipschitz, and convex. In other words, $L := \sup_{x \in \mathcal{X}} \sup_{\theta \in \Theta} \|\nabla_\theta \ell(\theta, x)\|_2$ is bounded.

(iii) $\pi$ is given by a restriction of $\tilde{\pi}$ to $\Theta$.

The following statements are the counterparts of Theorem 7 and its corollary.

**Theorem 10.** Let $\beta \in (0, 1]$ be a fixed parameter, and $D, D' \in \mathcal{X}^n$ be an adjacent pair of datasets. Under Assumption 9, inequality

$$G_{\beta,D} \left\{ \log \frac{dG_{\beta,D}}{dG_{\beta,D'}} \geq \varepsilon \right\} \leq \exp \left( -\frac{nm_\ell \beta + m_\pi}{4C'\beta^2} \left( \varepsilon - \frac{C'\beta^2}{nm_\ell \beta + m_\pi} \right)^2 \right) \tag{9}$$

holds for any $\varepsilon > \frac{C'\beta^2}{nm_\ell \beta + m_\pi}$. Here, we defined $C' := 2CL^2(1 + \log(\alpha^2/(\alpha^2 - 1)))$, where $C > 0$ is a universal constant that does not depend on any other quantities.

**Corollary 11.** Under Assumption 9, there exists an upper bound $B(\varepsilon, \delta) = B(\varepsilon, \delta, n, m_\ell, m_\pi, \alpha) > 0$, and $G_\beta(\theta \mid D_n)$ with $\beta \leq B(\varepsilon, \delta)$ satisfies $(\varepsilon, \delta)$-differential privacy.

Similar to Corollary 8, the upper bound on $\beta$ depends on the prior. Moreover, the right-hand side of (9) decreases to 0 as the size of dataset $n$ increases, which implies that $(\varepsilon, \delta)$-differential privacy is satisfied almost for free if the size of the dataset is large.

## 3.3 Example: Logistic regression

In this section, we provide an application of Theorem 7 to the problem of linear binary classification. Let $\mathcal{Z} := \{z \in \mathbb{R}^d, \|z\|_2 \leq r\}$ be a space of the input variables. The space of the observation is the set of input variables equipped with binary label $\mathcal{X} := \{x = (z, y) \in \mathcal{Z} \times \{-1, +1\}\}$. The problem is to determine a parameter $\theta = (a, b)$ of linear classifier $f_\theta(z) = \mathrm{sgn}(a^\top z + b)$.

Define a loss function $\ell_{\mathrm{LR}}$ by

$$\ell_{\mathrm{LR}}(\theta, x) := \log(1 + \exp(-y(a^\top z + b))). \tag{10}$$

The $\ell_2$-regularized logistic regression estimator is given by

$$\hat{\theta}_{\mathrm{LR}} = \operatorname*{argmin}_{\theta \in \mathbb{R}^{d+1}} \left\{ \frac{1}{n} \sum_{i=1}^n \ell_{\mathrm{LR}}(\theta, x_i) + \frac{\lambda}{2} \|\theta\|_2^2 \right\}, \tag{11}$$

where $\lambda > 0$ is a regularization parameter. Corresponding Gibbs posterior has a density

$$G_\beta(\theta \mid D) \propto \prod_{i=1}^n \sigma(y_i(a^\top z_i + b))^\beta \phi_{d+1}(\theta \mid 0, (n\lambda)^{-1}I), \tag{12}$$

where $\sigma(u) = (1 + \exp(-u))^{-1}$ is a sigmoid function, and $\phi_{d+1}(\theta \mid \mu, \Sigma)$ is a density of $(d+1)$-dimensional Gaussian distribution. It is easy to check that $\ell_{\mathrm{LR}(\cdot, x)}$ is $r$-Lipschitz and convex, and $-\log \phi_{d+1}(\cdot \mid 0, (n\lambda^{-1})I)$ is $(n\lambda)$-strongly convex. Hence, by Corollary 8, the Gibbs posterior satisfies $(\varepsilon, \delta)$-differential privacy if

$$\beta \leq \frac{\varepsilon}{2r} \sqrt{\frac{n\lambda}{1 + 2\log(1/\delta)}}. \tag{13}$$

# 4 Approximation Arguments

In practice, exact samplers of Gibbs posteriors (1) are rarely available. Actual implementations involve some approximation processes. Markov Chain Monte Carlo (MCMC) methods and Variational Bayes (VB) [1] are commonly used to obtain approximate samplers of Gibbs posteriors. The next proposition, which is easily obtained as a variant of Proposition 3 of [20], gives a differential privacy guarantee under approximation.

**Proposition 12.** Let $\rho : \mathcal{X}^n \to \mathcal{M}_+^1(\Theta)$ be a randomized estimator that satisfies $(\varepsilon, \delta)$-differential privacy. If for all $D$, there exist approximate sampling procedure $\rho'_D$ such that $d_{\mathrm{TV}}(\rho_D, \rho'_D) \leq \gamma$, then the randomized mechanism $D \mapsto \rho'_D$ satisfies $(\varepsilon\delta + (1 + e^{\varepsilon})\gamma)$-differential privacy. Here, $d_{\mathrm{TV}}(\mu, \nu) = \sup_{A \in \mathcal{T}} |\mu(A) - \nu(A)|$ is the total variation distance.

We now describe a concrete example of MCMC, the Langevin Monte Carlo (LMC). Let $\theta^{(0)} \in \mathbb{R}^d$ be an initial point of the Markov chain. The LMC algorithm for Gibbs posterior $G_{\beta, D}$ contains the following iterations:

$$\theta^{(t+1)} = \theta^{(t)} - h\nabla U(\theta^{(t)}) + \sqrt{2h}\eta^{(t+1)} \tag{14}$$

$$U(\theta) = \beta \sum_{i=1}^n \ell(\theta, x_i) - \log \pi(\theta). \tag{15}$$

Here $\eta^{(1)}, \eta^{(2)}, \ldots \in \mathbb{R}^d$ are noise vectors independently drawn from a centered Gaussian $N(0, I)$. This algorithm can be regarded as a discretization of a stochastic differential equation that has a stationary distribution $G_{\beta, D}$, and its convergence property has been studied in finite-time sense [9, 5, 11]. Let us denote by $\rho^{(t)}$ the law of $\theta^{(t)}$. If $d_{\mathrm{TV}}(\rho^{(t)}, G_{\beta, D}) \leq \gamma$ holds for all $t \geq T$, then the privacy of the LMC sampler is obtained from Proposition 12. In fact, we can prove by Corollary 1 of [9] the following proposition.

**Proposition 13.** Assume that Assumption 6 holds. Let $\ell(\theta, x)$ be $M_\ell$-smooth for all $x \in \mathcal{X}$, and $-\log \pi(\theta)$ be $M_\pi$-smooth. Let $d \geq 2$ and $\gamma \in (0, 1/2)$. We can choose $\beta > 0$, by Corollary 8, so that $G_{\beta, D}$ satisfies $(\varepsilon, \delta)$-differential privacy. Let us set step size $h$ of the LMC algorithm (14) as

$$h = \frac{2m_\pi\gamma^2}{d(n\beta M_\ell + M_\pi)^2 \left[4\log\left(\frac{1}{\gamma}\right) + d\log\left(\frac{n\beta M_\ell + M_\pi}{m_\pi}\right)\right]}, \tag{16}$$

and set $T$ as

$$T = \frac{d(n\beta M_\ell + M_\pi)^2}{4m_\pi\gamma^2} \left[4\log\left(\frac{1}{\gamma}\right) + d\log\left(\frac{n\beta M_\ell + M_\pi}{m_\pi}\right)\right]^2. \tag{17}$$

Then, after $T$ iterations of (14), $\theta^{(T)}$ satisfies $(\varepsilon, \delta + (1 + e^{\varepsilon})\gamma)$-differential privacy.

The algorithm suggested in Proposition 13 is closely related to the differentially private stochastic gradient Langevin dynamics (DP-SGLD) proposed by Wang, et al. [20]. Ignoring the computational cost, we can take the approximation error level $\gamma > 0$ arbitrarily small, while the convergence property to the target posterior distribution is not necessarily ensured about DP-SGLD.

# 5 Proofs

In this section, we give a formal proof of Theorem 7 and a proof sketch of 10.

There is a vast literature on techniques to obtain a concentration inequality in (5) (see, for example, [4]). Logarithmic Sobolev inequality (LSI) is a useful tool for this purpose. We say that a probability measure $\mu$ over $\Theta \subset \mathbb{R}^d$ satisfies LSI with constant $D_{\mathrm{LS}}$ if inequality

$$\mathbb{E}_\mu[f^2 \log f^2] - \mathbb{E}_\mu[f^2] \log \mathbb{E}_\mu[f^2] \leq 2D_{\mathrm{LS}}\mathbb{E}_\mu \|\nabla f\|_2^2 \tag{18}$$

holds for any integrable function $f$, provided the expectations in the expression are defined. It is known that [15, 4], if $\mu$ satisfies LSI, then every real-valued $L$-Lipschitz function $F$ behaves in a sub-Gaussian manner:

$$\mu\{F \geq \mathbb{E}_\mu[F] + t\} \leq \exp\left(-\frac{t^2}{2L^2 D_{\mathrm{LS}}}\right). \tag{19}$$

In our analysis, we utilize the LSI technique for the following two reasons: (a) a sub-Gaussian tail bound of the log-density ratio is obtained from (19), and (b) an upper bound on the KL-divergence is directly obtained from LSI, which appears to be difficult to prove by any other argument.

Roughly speaking, LSI holds if the logarithm of the density is strongly concave. In particular, for a Gibbs measure on $\mathbb{R}^d$, the following fact is known.

**Lemma 14** ([15]). Let $U : \mathbb{R}^d \to \mathbb{R}$ be a twice differential, $m$-strongly convex and integrable function. Let $\mu$ be a probability measure on $\mathbb{R}^d$ whose density is proportional to $\exp(-U)$. Then $\mu$ satisfies LSI (18) with constant $D_{\mathrm{LS}} = m^{-1}$.

In this context, the strong convexity of $U$ is related to the curvature-dimension condition $\mathrm{CD}(m, \infty)$, which can be used to prove LSI on general Riemannian manifolds [19, 2].

*Proof of Theorem 7.* For simplicity, we assume that $\ell(\cdot, x)$ ($\forall x \in \mathcal{X}$) is twice differentiable. For general Lipschitz and convex loss functions (e.g. hinge loss), the theorem can be proved using a mollifier argument. Since $U(\cdot) = \beta \sum_i \ell(\cdot, x_i) - \log \pi(\cdot)$ is $m_\pi$-strongly convex, Gibbs posterior $G_{\beta,D}$ satisfies LSI with constant $m_\pi^{-1}$.

Let $D, D' \in \mathcal{X}^n$ be a pair of adjacent datasets. Considering appropriate permutation of the elements, we can assume that $D = (x_1, \ldots, x_n)$ and $D' = (x'_1, \ldots, x'_n)$ differ in the first element, namely, $x_1 \neq x'_1$ and $x_i = x'_i$ ($i = 2, \ldots, n$). By the assumption that $\ell(\cdot, x)$ is $L$-Lipschitz, we have

$$\left\| \nabla \log \frac{\mathrm{d}G_{\beta,D}}{\mathrm{d}G_{\beta,D'}} \right\|_2 = \beta \| \nabla(\ell(\theta, x_1) - \ell(\theta, x'_1)) \|_2 \leq 2\beta L, \tag{20}$$

and log-density ratio $\log \frac{\mathrm{d}G_{\beta,D}}{\mathrm{d}G_{\beta,D'}}$ is $2\beta L$-Lipschitz. Then, by concentration inequality for Lipschitz function (19), we have

$$\forall t > 0, \quad G_{\beta,D} \left\{ \log \frac{\mathrm{d}G_{\beta,D}}{\mathrm{d}G_{\beta,D'}} \geq D_{\mathrm{KL}}(G_{\beta,D}, G_{\beta,D'}) + t \right\} \leq \exp\left( -\frac{m_\pi t^2}{8 L^2 \beta^2} \right) \tag{21}$$

We will show an upper bound of the KL-divergence. To simplify the notation, we will write $F := \frac{\mathrm{d}G_{\beta,D}}{\mathrm{d}G_{\beta,D'}}$. Noting that

$$\| \nabla \sqrt{F} \|_2^2 = \| \nabla \exp(2^{-1} \log F) \|_2^2 = \left\| \frac{\sqrt{F}}{2} \nabla \log F \right\|_2^2 \leq \frac{F}{4} \cdot (2\beta L)^2 \tag{22}$$

and that

$$D_{\mathrm{KL}}(G_{\beta,D}, G_{\beta,D'}) = \mathbb{E}_{G_{\beta,D}}[\log F]$$
$$= \mathbb{E}_{G_{\beta,D'}}[F \log F] - \mathbb{E}_{G_{\beta,D'}}[F] \mathbb{E}_{G_{\beta,D'}}[\log F], \tag{23}$$

we have, from LSI (18) with $f = \sqrt{F}$,

$$D_{\mathrm{KL}}(G_{\beta,D}, G_{\beta,D'}) \leq \frac{2}{m_\pi} \mathbb{E}_{G_{\beta,D'}} \| \nabla \sqrt{F} \|_2^2 \leq \frac{2 L^2 \beta^2}{m_\pi} \mathbb{E}_{G_{\beta,D'}}[F] = \frac{2 L^2 \beta^2}{m_\pi}. \tag{24}$$

Combining (21) and (24), we have

$$G_{\beta,D} \left\{ \log \frac{\mathrm{d}G_{\beta,D}}{\mathrm{d}G_{\beta,D'}} \geq \varepsilon \right\} \leq G_{\beta,D} \left\{ \log \frac{\mathrm{d}G_{\beta,D}}{\mathrm{d}G_{\beta,D'}} \geq \varepsilon + D_{\mathrm{KL}}(G_{\beta,D}, G_{\beta,D'}) - \frac{2 L^2 \beta^2}{m_\pi} \right\}$$
$$\leq \exp\left( -\frac{m_\pi}{8 L^2 \beta^2} \left( \varepsilon - \frac{2 L^2 \beta^2}{m_\pi} \right)^2 \right) \tag{25}$$

for any $\varepsilon > \frac{2 L^2 \beta^2}{m_\pi}$. $\qquad\qquad\square$

*Proof sketch for Theorem 10.* The proof is almost the same as that of Theorem 7. It is sufficient to show that the set of Gibbs posteriors $\{G_{\beta,D}, \ D \in \mathcal{X}^n\}$ simultaneously satisfies LSI with the same constant. Since the logarithm of the density is $m := (n m_\ell \beta + m_\pi)$-strongly convex, a probability measure $\tilde{G}_{\beta,D}$ satisfies LSI with constant $m^{-1}$. By the Poincaré inequality for $\tilde{G}_{\beta,D}$, the variance of $\|\theta\|_2$ is bounded by $d/m \leq d/m_\pi$. By the Chebyshev inequality, we can check that the mass of parameter space is lower-bounded as $\tilde{G}_{\beta,D}(\Theta) \geq p := 1 - \alpha^{-2}$. Then, by Corollary 3.9 of [17], $G_{\beta,D} := \tilde{G}_{\beta,D}|_\Theta$ satisfies LSI with constant $C(1 + \log p^{-1}) m^{-1}$, where $C > 0$ is a universal numeric constant. $\qquad\qquad\square$

**Acknowledgments**

This work was supported by JSPS KAKENHI Grant Number JP15H02700.

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
