[Supplementary Material]

# A  More on the exponential mechanism

It is convenient to understand the exponential mechanism as a composition of two Lipschitz maps. We define a distance $d_{\mathrm{DP}}$ between two probability measures $\mu_1, \mu_2 \in \mathcal{M}_+^1(\Theta)$ by

$$d_{\mathrm{DP}}(\mu_1, \mu_2) := \sup_{A \subset \Theta} |\log \mu_1(A) - \log \mu_2(A)|, \tag{A.1}$$

where the supremum is taken over measurable sets. $d_{\mathrm{DP}}(\mu_1, \mu_2)$ is defined to be $+\infty$ if $\mu_1$ and $\mu_2$ are not absolutely continuous. It is easy to check that the $(\varepsilon, 0)$-differential privacy of randomized estimator $\rho$ is equivalent to the $\varepsilon$-Lipschitz property as a map between two metric spaces $\rho : \mathcal{X}^n \to \mathcal{M}_+^1(\Theta)$. We define a function space $\mathbb{R}^\Theta := \{f : \Theta \to \mathbb{R}\}$ equipped with supremum distance $d_\infty(f, g) := \sup_\theta |f(\theta) - g(\theta)|$. If the sensitivity $\Delta_{\mathcal{L}}$ is finite, a function-valued function $\mathcal{L} : D_n \mapsto \mathcal{L}(\cdot, D_n)$ is $\Delta_{\mathcal{L}}$-Lipschitz with respect to $d_H$ and $d_\infty$. We define a Gibbs map $G_\beta : \mathbb{R}^\Theta \to \mathcal{M}_+^1(\Theta)$ as follows: given a function $f$, $G_\beta(f)$ is a probability distribution whose density w.r.t. $\pi$ is proportional to $\exp(-\beta f)$. We can check that the Gibbs map is $2\beta$-Lipschitz. Eventually, Theorem 3 states that the exponential mechanism is $2\beta\Delta_{\mathcal{L}}$-Lipschitz, because it is a composition of two Lipschitz functions:

$$(\mathcal{X}^n, d_H) \xrightarrow{\mathcal{L}} (\mathbb{R}^\Theta, d_\infty) \xrightarrow{G_\beta} (\mathcal{M}_+^1(\Theta), d_{\mathrm{DP}}). \tag{A.2}$$

# B  Gaussian mean estimation

In Section 3.2, $(\varepsilon, \delta)$-differential privacy of Gibbs posteriors with strongly convex loss functions were investigated. A typical example of strongly convex loss function is a *quadratic loss* $\ell(\theta, x) = \frac{1}{2} \|\theta - x\|_2^2$, which arises in many fields in statistics and machine learning. Theorem 10 can be applied if we use a restricted Gaussian prior, and the resulting Gibbs posterior satisfies $(\varepsilon, \delta)$-differential privacy. In this case, however, a tighter evaluation can be obtained because the posterior becomes a Gaussian distribution and the KL-divergence is calculated in closed form.

We provide a motivating example of *Gaussian mean estimation*. Let $X$ be a bounded random variable in $\mathbb{R}^d$ that satisfies $\|X\|_2 \leq r$ with a known constant $r > 0$. Observing an i.i.d. sample $D = \{x_1, \ldots, x_n\}$, we consider the problem of estimating the mean $\theta_0 = \mathbb{E}[X]$. The classical least square estimator that minimizes the empirical risk $\frac{1}{2} \sum_{i=1}^n \|\theta - x_i\|_2^2$ is given by the sample mean $\bar{x} = \frac{1}{n} \sum_{i=1}^n x_i$. Here, we consider a (quasi-)Bayesian conterpert of the least square estimator. Let $\pi(\theta)$ be an isotropic Gaussian prior on $\theta$:

$$\pi(\theta) = \frac{1}{\sqrt{2\pi}^d} \exp\left(-\frac{\lambda}{2} \|\theta\|_2^2\right). \tag{B.1}$$

Then, the Gibbs posterior

$$G_\beta(\theta|D) \propto \exp\left(-\frac{\beta}{2} \sum_{i=1}^n \|\theta - x_i\|_2^2\right) \pi(\theta) \tag{B.2}$$

is a $d$-dimensional Gaussian distribution $N_d(\bar{\mu}, \bar{\sigma}^2 I_d)$, in which the mean $\bar{\mu}$ and the variance $\bar{\sigma}^2$ are given by

$$\bar{\mu} = \frac{n\beta}{n\beta + \lambda} \bar{x}, \quad \bar{\sigma}^2 = \frac{1}{n\beta + \lambda}. \tag{B.3}$$

The Gibbs posterior satisfies $(\varepsilon, \delta)$-differential privacy if $\beta > 0$ is taken sufficiently small. In fact, we have the following theorem.

**Theorem 15.** Let $\beta \in (0, 1]$ be a fixed parameter, and $D, D' \in \mathcal{X}^n$ be an adjacent pair of datasets. Then the Gaussian posterior defined in (B.2) satisfies

$$G_{\beta,D}\left\{\log \frac{\mathrm{d}G_{\beta,D}}{\mathrm{d}G_{\beta,D'}} \geq \varepsilon\right\} \leq \exp\left(-\frac{n\beta + \lambda}{8r^2\beta^2}\left(\varepsilon - \frac{2r^2\beta^2}{n\beta + \lambda}\right)^2\right) \tag{B.4}$$

for $\varepsilon \geq \frac{2\beta^2 r^2}{\beta n + \lambda}$. Especially, the Gibbs posterior satisfies $(\varepsilon, \delta)$-differential privacy with $\delta$ given by the right-hand side of (B.4).

*Proof.* Assume that $D$ and $D'$ differs in the first element. $G_{\beta,D}$ is a Gaussian distribiton and satisfies LSI with constant $\bar{\sigma}^{-1} = n\beta + \lambda$. Thus, it suffices to prove a bound of the KL-divergence $D_{\mathrm{KL}}(G_{\beta,D}, G_{\beta,D'})$. It is well known that the KL-divergence between two Gaussian distributions is given by

$$
D_{\mathrm{KL}}(N_d(\mu_1, \Sigma_1), N_d(\mu_2, \Sigma_2))
$$
$$
= \frac{1}{2} \left\{ \mathrm{Tr}\Sigma_2^{-1}\Sigma_1 + (\mu_2 - \mu_1)^\top \Sigma_2^{-1}(\mu_2 - \mu_1) - d + \log \frac{|\Sigma_2|}{|\Sigma_1|} \right\}. \tag{B.5}
$$

Applying this formula with $\mu_1 = \frac{\beta}{n\beta+\lambda} \sum_{x \in D} x$, $\mu_2 = \frac{\beta}{n\beta+\lambda} \sum_{x \in D'} x$, and $\Sigma_1 = \Sigma_2 = \frac{1}{n\beta+\lambda}I_d$, we have

$$
D_{\mathrm{KL}}(G_{\beta,D}, G_{\beta,D'}) = \frac{\beta^2}{2(n\beta + \lambda)} \|x_1 - x_1'\|_2^2 \leq \frac{2\beta^2 r^2}{\beta n + \lambda}. \tag{B.6}
$$

Combining this with the concentration inequality (19), we get the desired result. $\square$

We provide some comments on the effectiveness of the upper bound (B.4), compared to the exponential mechanism and the Gaussian mechanism. If $(\varepsilon, \delta) = (0.1, 0.001)$ and $R = 1$ and $\lambda = 0$, we can check from (B.4) that the Gaussian posterior satisfies $(\varepsilon, \delta)$-differential privacy for $\beta < 1.79 \times 10^{-4} n$. In contrast, if we use a prior supported on a compact ball $|\theta| \leq 1$, the exponential mechanism suggests that $\beta \leq \varepsilon/8 = 0.0125$ works. Then, the $(\varepsilon, \delta)$-type analysis yields a larger upper bound if $n \geq 15$. Moreover, if $n > 5600$, the (pseudo-)Bayesian posterior (i.e. $\beta = 1$) automatically satisfies $(0.1, 0.001)$-differential privacy.

If $\pi$ is the uniform prior (i.e. $\lambda = 0$), the Gaussian posterior (B.2) can be regarded as the Gaussian mechanism. The Gaussian mechanism suggests that taking

$$
\beta \leq \frac{\varepsilon^2 n}{8r^2 \log \frac{2}{\delta}} \tag{B.7}
$$

is sufficient to $(\varepsilon, \delta)$-differential privacy (e.g. Proposition 1.3.3 of [5]). For example, this upper bound becomes $1.26 \times 10^{-4} n$ for $(\varepsilon, \delta) = (0.1, 0.001)$. On the other hand, (B.4) is satisfied if

$$
\beta \leq \frac{n}{2r^2}\eta(\varepsilon, \delta), \tag{B.8}
$$

where

$$
\eta(\varepsilon, \delta) = \varepsilon + 2\log(\delta^{-1}) - 2\sqrt{\log(\delta^{-1})(\varepsilon + \log(\delta^{-1}))}. \tag{B.9}
$$

We now compare the upper bounds (B.7) and (B.8) when $a = \frac{\varepsilon}{\log(\delta^{-1})}$ is sufficiently small. Using Taylor expansion of $g(a) = 1 + \frac{2}{a} - 2\sqrt{\frac{1}{a}\left(1 + \frac{1}{a}\right)}$ around $a = 0$, we can approximate $\eta(\varepsilon, \delta)$ as

$$
\eta(\varepsilon, \delta) = \varepsilon \left[ \frac{\varepsilon}{4\log(\delta^{-1})} + O\left( \frac{\varepsilon^2}{(\log(\delta^{-1}))^2} \right) \right].
$$

Hence, the right-hand side of (B.8) has the leading term $\varepsilon^2 n/8r^2 \log \frac{2}{\delta}$, which coincides with (B.7).

## C   Utility analysis

In this section, we provide some utility analyses for the Gibbs posterior sampling algorithm.

### C.1   Consistency in parametric statistics

Let $\mathcal{P} = \{p_\theta = p(\cdot \mid \theta) \mid \theta \in \Theta\}$ be a statistical model, which is a family of probability densities indexed by $\Theta \subset \mathbb{R}^d$. In parametric statistics, a natural loss function is the negative log-likelihood function $\ell(\theta, x) = -\log p(x \mid \theta)$. In this case, the density of the Gibbs posterior is given by $G_\beta(\theta \mid D) \propto \pi(\theta) \prod_{i=1}^n p(x_i \mid \theta)^\beta$. Given an i.i.d. sample $D = D_n = \{x_1, \ldots, x_n\}$ that is drawn from a distribution in $\mathcal{P}$, we consider the problem of estimating the true parameter $\theta_0 \in \Theta$. For this setting, it is already shown in [9] that the Gibbs posterior is consistent if the pair $(\mathcal{P}, \pi)$ satisfies the usual regularity conditions for the consistency. In this subsection, we provide similar consistency results for reader's convenience.

We consider the case that the assumption of our main theorem (Theorem 7) is satisfied. By Corollary 8, the Gibbs posterior satisfies $(\varepsilon, \delta)$-differential privacy if $\beta < B(\varepsilon, \delta)$, where

$$B(\varepsilon, \delta) = \frac{\varepsilon}{2L}\sqrt{\frac{m_\pi}{1 + 2\log(1/\delta)}}. \tag{C.1}$$

For simplicity, we assume that $\beta$ is a fixed value and does not depend on the sample size $n$.

**Theorem 16.** Let $D = \{x_1, \ldots, x_n\}$ be an i.i.d. sample from a distribution $p_0 = p(\cdot \mid \theta_0)$. Assume that Assumption 6 is satisfied for $\ell(\theta, x) = -\log p(x \mid \theta)$ and $\pi(\theta)$. We also assume the following conditions:

 (i) $\pi(\theta_0) > 0$,

 (ii) the function $\theta \mapsto \log p(X \mid \theta)$ is differentiable at $\theta_0 \in \Theta$ in $p_0$-almost surely,

 (iii) the Fisher information matrix $I_0 = I(\theta_0) = -\mathbb{E}_{\theta_0}[\nabla^2 \ell]$ is invertible, and

 (iv) there exists a uniformly consistent estimator $T_n = T_n(D_n)$ for $\theta \in \Theta$, i.e.

$$\sup_{\theta \in \Theta} \Pr_\theta \{\|T_n - \theta\| \geq \varepsilon\} \to 0.$$

Then, we have the following statements.

 1. (Consistency in the Frequentist sense) For $n \geq 1$, let $\hat{\theta}_n \sim G_\beta(\cdot \mid D_n)$ be the output of the Gibbs posterior sampling algorithm. Then, $\|\hat{\theta}_n - \theta_0\|_2 \to 0$ holds in $p_0$-probability.

 2. (Posterior convergence rate) For any sequence $M_n \to \infty$, we have

$$G_{\beta, D_n}\left\{\|\theta - \theta_0\| \geq M_n/\sqrt{n}\right\} \xrightarrow{p_0} 0. \tag{C.2}$$

 3. (Misspecified Bernstein–von Mises theorem) Let $\mathcal{N}_{\mu, \Sigma}(A)$ be the probability of an event $A \subset \mathbb{R}^d$ with respect to the distribution $N_d(\mu, \Sigma)$. Then the Gibbs posterior converges to a normal distributions with mean $\theta_0$ and covariance matrix $(n\beta)^{-1}I_0^{-1}$:

$$\sup_{B \subset \Theta}\left|G_{\beta, D_n}\{\sqrt{n}(\theta - \theta_0) \in B\} - \mathcal{N}_{0, \beta^{-1}I_0^{-1}}(B)\right| \xrightarrow{p_0} 0. \tag{C.3}$$

*Proof.* Since (C.2) implies the consistency, it suffice to show the second and third statements. The Gibbs posterior $G_{\beta, D}$ can be regarded as a Bayesian posterior in a (improper) misspecified model $\mathcal{P}^\beta = \{p(\cdot \mid \theta)^\beta \mid \theta \in \Theta\}$. The following proof is a direct application of the general theory for posterior contraction in misspecified model [6]. We should note that, although elements in $\mathcal{P}^\beta$ are not probability densities, the results in [6] can be applied.

Note that any element of $\mathcal{P}^\beta$ can be written as $q = p_\theta^\beta$ with a$\theta \in \Theta$. Since the true parameter is included in $\Theta$, the (non-normalized) KL-divergence $\mathbb{E}_{p_0} \log \frac{p_0^\beta}{q} = \beta D_{\mathrm{KL}}(p_0, p_\theta)$ defined on $\mathcal{P}^\beta$ is minimized by $q_0 = p_0^\beta$. Hence, we want to prove the contraction of the Gibbs posterior around $\theta_0$. In fact, the assumptions of Theorem 3.1 of [6] are easily checked, and thus we have (C.2). Note that (iv) implies the existence of a suitable sequence of tests that distinguishes $\theta_0$ and slightly distant parameter ([8], Section 10.2).

To prove the normal approximation (C.3), we require an additional condition in which the log-likelihood ratio is well-approximated by a quadratic form around $\theta_0$. By Lemma 2.1 of [6], the model $\mathcal{P}$ satisfies the local asymptotic normality (LAN) condition around the true parameter $\theta_0 \in \Theta$. More precisely, for every compact set $H \subset \mathbb{R}^d$,

$$\sup_{h \in H}\left|\sum_{i=1}^n [\log p(x_i \mid \theta_0 + h/\sqrt{n}) - \log p(x_i \mid \theta_0)] - \frac{\beta}{2}h^\top I_0 h\right| \to 0$$

holds in outer $p_0$-probability. Then, by Theorem 2.1 of [6], we have (C.3). □

In the above proposition, condition (iv) can be replaced by the existence of a suitable sequence of test, albeit with a more complex statement. Such conditions can be checked from identifiability of the model and some continuity conditions (e.g. Lemma 10.4 or Lemma 10.6 in [8]).

**Example 17** (Bernoulli distribution with logit-normal prior). Let $\tilde{p}(x \mid p) = p^x(1-p)^{1-x}$ ($x \in \{0,1\}, p \in (0,1)$) be the likelihood function of the Bernoulli distribution. In this case, the map $p \mapsto -\log \tilde{p}(x \mid p)$ is not Lipschitz nor convex. However, reparametrizing as $\theta = \sigma^{-1}(p) = \log \frac{p}{1-p}$, $\ell(\theta, x) = -\log \tilde{p}(x \mid \sigma(\theta)) = -x\theta + \log(1 + e^\theta)$ is 1-Lipschitz and convex for $x \in \{0,1\}$.

We now consider a normal prior $\pi(\theta) = \phi_1(\theta \mid \mu, v)$. When $\theta \sim \pi(\theta)$, the distribution of $p = \sigma(\theta)$ is called a "logit-normal" distribution. Given any dataset $D \in \{0,1\}^n$, the Gibbs posterior is written as $G_\beta(\theta \mid D) \propto \pi(\theta)\sigma(\theta)^{n\beta\bar{x}}(1-\sigma(\theta))^{n\beta(1-\bar{x})}$, where $\bar{x} = n^{-1}\sum_i x_i$ is the maximum likelihood estimator for $p$. Since $-\log \pi(\theta)$ is $v^{-1}$-strongly convex, we can apply Theorem 7. In fact, the Gibbs posterior satisfies $(\varepsilon, \delta)$-differential privacy if $\beta \leq 2^{-1}\varepsilon(v(1 + 2\log(\delta^{-1})))^{-1/2}$. For example, if $(\varepsilon, \delta) = (0.1, 0.001)$ and $v = 1$, it is satisfied with $\beta = 0.012$.

We can also prove that there exists a uniformly consistent estimator for this model. In fact, since the above statistical model is identifiable and the map $\theta \mapsto p_\theta = \tilde{p}(\cdot \mid \sigma(\theta))$ is continuous in total-variation distance[1], we can apply Theorem 10.6 in [8]. Consequently, the Gibbs posterior is consistent, and converges to a normal distribution. The asymptotic variance is given by $[\beta\sigma(\theta_0)(1 - \sigma(\theta_0))]^{-1}$, which is at least 83 times bigger than the Bayesian case.

## C.2 PAC-Bayesian bounds

In this subsection, we study the relationship between the Gibbs posteriors and some distribution-dependent risk bounds, namely the PAC-Bayesian bounds. Define the risk $R(\theta)$ and the empirical risk $R_n(\theta)$ by

$$R(\theta) = \mathbb{E}_P[\ell(\theta, X)], \qquad R_n(\theta) = \frac{1}{n}\sum_{i=1}^n \ell(\theta, x_i),$$

where $\mathbb{E}_P$ is the expectation with respect to an unknown probability measure $P$. It is well-known that the Gibbs posterior can be characterized as the optimal random estimator for PAC-Bayesian learning (e.g. [3, 1]). In particular, the Gibbs posterior minimizes the following quantity that commonly appears in the PAC-Bayesian upper bounds:

$$G_{\beta, D} \in \underset{\rho \in \mathcal{M}_+^1(\Theta)}{\operatorname{argmin}} \left\{ \mathbb{E}_\rho R_n(\theta) + \frac{1}{\beta} D_{\mathrm{KL}}(\rho, \pi) \right\}. \tag{C.4}$$

Therefore, these risk bounds can be useful for analysing the generalization performance of the differentially private posterior sampling algorithms. To the best of our knowledge, the relationship between the exponential mechanism and the PAC-Bayesian bound was first pointed out by Mir (2012) [7]. Here, we provide an example of PAC-Bayesian upper bound, which is applicable to the Gibbs posterior discussed in Section 3. The following theorem is proved in Theorem 4.1 and Theorem 4.2 in [1]

**Theorem 18** ([1]). We say that a Hoeffding assumption is satisfied for prior $\pi$ when there is a function $f(\beta, n)$ and an interval $\beta^* > 0$ such that, for any $\beta \in (0, \beta^*)$ and $\theta \in \Theta$,

$$\left. \begin{array}{l} \mathbb{E}_\pi \mathbb{E}_{P^n}[\exp(\beta(R(\theta) - R_n(\theta)))] \\ \mathbb{E}_\pi \mathbb{E}_{P^n}[\exp(\beta(R_n(\theta) - R(\theta)))] \end{array} \right\} \leq \exp(f(\beta, n)). \tag{C.5}$$

Assume that Assumption 6 and Hoeffding assumption hold. Then, for any $\beta \in (0, \beta^*)$ and $\alpha \in (0, 1)$,

$$\mathbb{E}_{G_{\beta, D}} R(\theta) \leq \mathbb{E}_{G_{\beta, D}} R_n(\theta) + \frac{f(\beta, n) + D_{\mathrm{KL}}(G_{\beta, D}, \pi) + \log \alpha^{-1}}{\beta} \tag{C.6}$$

holds with probability at least $1-\alpha$. Moreover, we have an oracle-type inequality: for any $\alpha \in (0,1)$, we have

$$\mathbb{E}_{G_{\beta, D}} R(\theta) \leq \inf_{\rho \in \mathcal{M}_+^1(\Theta)} \left\{ \mathbb{E}_\rho R(\theta) + 2\frac{f(\beta, n) + D_{\mathrm{KL}}(\rho, \pi) + \log \alpha^{-1}}{\beta} \right\} \tag{C.7}$$

with probability at least $1 - 2\alpha$.

**Example 19** (Hinge loss). Let $\mathcal{X} = \mathcal{Z} \times \{-1, +1\}$ be a binary-labeled feature space which we defined in Section 3.3. To simplify the notation, we omit the bias term $b$. Here, we consider the hinge loss $\ell_{\text{hinge}} : \mathbb{R}^d \times \mathcal{X} \to \mathbb{R}$ defined by

$$\ell_{\text{H}}(\theta, x) = \max\{0, 1 - y\theta^\top z\}.$$

We use a Gaussian prior $\pi(\theta) = \phi_d(\theta \mid 0, (n\lambda)^{-1}I)$. Then, by Lemma 6.1 in [1], Hoeffding assumption (C.5) is satisfied with $f(\beta, n) = \beta^2/4n - \frac{1}{2}\log\left(1 - \frac{r^2\beta^2}{4\lambda n^2}\right)$ and $\beta^* = 2\sqrt{\lambda}n/r$. By Corollary 8 and Theorem 18, the Gibbs posterior $G_{\beta,D}$ simultaneously satisfies $(\varepsilon, \delta)$-differential privacy and the risk bounds (C.6) (C.7) if

$$\beta \leq \frac{\sqrt{\lambda}n}{r} \min\left\{\frac{\varepsilon}{2\sqrt{1 + 2\log(\delta^{-1})}}, 2\sqrt{n}\right\}.$$

# D   Omitted proofs

## D.1   Proofs in Section 3.1

In this subsection, we provide formal proofs for Theorem 7. In the simplified proof in Section 5, we assumed that $\ell(\cdot, x)$ is twice differentiable. However, Theorem 7 still holds for non-differentiable loss functions such as the hinge losses. Note that a function $f : \mathbb{R}^d \to \mathbb{R}$ is said to be $m$-strongly convex, if it satisfies

$$f(\lambda x + (1 - \lambda)y) \leq \lambda f(x) + (1 - \lambda)f(y) - \frac{m}{2}\lambda(1 - \lambda)\|x - y\|_2^2 \tag{D.1}$$

for all $x, y \in \mathbb{R}^d$ and $t \in [0, 1]$. If $f$ is twice differentiable, this is equivelent to the definition given in Section 1.2.

Let $\varphi : \mathbb{R}^d \to \mathbb{R}$ be a *mollifier*, namely, a compactly supported $C^\infty$-function that satisfies (a) $\int_{\mathbb{R}^d} \varphi(x)\mathrm{d}x = 1$ and (b) $\lim_{a\downarrow 0} a^{-d}\varphi(a^{-1}x) = \delta(x)$ (the convergence is understood in the space of Schwartz distributions). For example, we can take $\varphi$ as

$$\varphi(x) = \begin{cases} Z^{-1}\exp\left(-\dfrac{1}{1 - \|x\|_2^2}\right) & \text{if } \|x\|_2^2 \leq 1 \\ 0 & \text{otherwise} \end{cases}, \tag{D.2}$$

where $Z$ is a normalization constant. For any $f : \mathbb{R}^d \to \mathbb{R}$ and $a > 0$, define a new function $\Phi_a(f) : \mathbb{R}^d \to \mathbb{R}$ by

$$\Phi_a(f)(x) = \int_{\mathbb{R}^d} a^{-d}\varphi(a^{-1}(x - y))f(y)\mathrm{d}y$$

$$= \int_{\mathbb{R}^d} a^{-d}\varphi(a^{-1}x)f(x - y)\mathrm{d}y. \tag{D.3}$$

Then $\Phi_a(f)$ becomes a $C^\infty$-function and converges pointwise to $f$ as $a \downarrow 0$. Furthermore, $\Phi_a$ preserves the convexity in the following sense.

**Lemma 20.** If $f : \mathbb{R}^d \to \mathbb{R}$ is a convex function, then $\Phi_a(f)$ is $C^\infty$ and convex. Moreover, if $f$ is $m$-strongly convex, then $\Phi_a(f)$ is also $m$-strongly convex.

*Proof.* We will prove the latter statement. Since $f$ is $m$-strongly convex, we have

$$\Phi_a(f)(\lambda x + (1 - \lambda)y) = \int_{\mathbb{R}^d} a^{-d}\varphi(a^{-1}z)f(\lambda x + (1 - \lambda)y)\mathrm{d}z$$

$$\leq \int_{\mathbb{R}^d} a^{-d}\varphi(a^{-1}z)\left[\lambda f(x - z) + (1 - \lambda)f(y - z) - \frac{m}{2}\lambda(1 - \lambda)\|x - y\|_2^2\right]\mathrm{d}z$$

$$= \lambda\Phi_a(f)(x) + (1 - \lambda)\Phi_a(f)(y) - \frac{m}{2}\lambda(1 - \lambda)\|x - y\|_2^2.$$

for all $x, y \in \mathbb{R}^d$ and $\lambda \in [0, 1]$. $\qquad\square$

We are ready to prove Theorem 7.

*Proof of Theorem 7.* We will write $\mathcal{L}(\theta) = \beta \sum_{i=1}^{n} \ell(\theta, x_i)$. By Lemma 20, $\Phi_a(\mathcal{L})$ is an infinitely diferentiable convex function for any $a > 0$. Define a probability measure $G_{\beta,D}^a$ with a density

$$G_\beta^a(\theta|D) = \frac{\exp(-\Phi_a(\mathcal{L})(\theta))\pi(\theta)}{\int_{\mathbb{R}^d} \exp(-\Phi_a(\mathcal{L})(\theta))\pi(\theta)\mathrm{d}\theta}. \tag{D.4}$$

Since $\ell(\theta, x)$ is assumed to be non-neggative, $\mathcal{L}$ and $\Phi_a(\mathcal{L})$ are both non-negattive by their definition. Hence $\exp(-\Phi_a(\mathcal{L})(\theta)) \leq 1$, and by the dominated convergence theorem we have

$$\int_{\mathbb{R}^d} \exp(-\Phi_a(\mathcal{L})(\theta))\pi(\theta)\mathrm{d}\theta \to \int_{\mathbb{R}^d} \exp(-\mathcal{L}(\theta))\pi(\theta)\mathrm{d}\theta \tag{D.5}$$

as $a \downarrow 0$. Therefore, $G_\beta^a(\theta|D)$ converges pointwise to $G_\beta(\theta|D)$ as $a \downarrow 0$, and $G_{\beta,D}^a$ converges weakly to the Gibbs posterior $G_{\beta,D}$ ([2], p.29).

Since $U_a(\theta) = \beta\Phi_a(\mathcal{L})(\theta) - \log\pi(\theta)$ is $m_\pi$-strongly convex, $G_{\beta,D}^a$ satisfies LSI with constant $m_\pi^{-1}$. By a similar argument to Section 5, we have

$$\mathbb{E}_{G_{\beta,D}^a}\left[\log\frac{\mathrm{d}G_{\beta,D}}{\mathrm{d}G_{\beta,D'}}\right] \leq \frac{2L^2\beta^2}{m_\pi} \tag{D.6}$$

and

$$G_{\beta,D}^a\left\{\log\frac{\mathrm{d}G_{\beta,D}}{\mathrm{d}G_{\beta,D'}} \geq \varepsilon\right\} \leq \exp\left(-\frac{m_\pi}{8L^2\beta^2}\left(\varepsilon - \frac{2L^2\beta^2}{m_\pi}\right)^2\right) \tag{D.7}$$

with $\varepsilon > \frac{2L^2\beta^2}{m_\pi}$. Since the set $\{\theta \in \mathbb{R}^d : \log\frac{\mathrm{d}G_{\beta,D}}{\mathrm{d}G_{\beta,D'}} \geq \varepsilon\}$ is obviously a continuity set (i.e. having no mass on its boundary), taking the limit as $a \downarrow 0$ in the both side of (D.7) yields the desired result. $\square$

## D.2 Proofs in Section 4

*Proof of Proposition 12.* Let $D, D' \in \mathcal{X}^n$ be an adjacent pair of datasets. For any measurable set $A \subset \Theta$, we have

$$\begin{aligned}\rho'_D(A) &\leq \rho_D(A) + \gamma \leq e^\varepsilon \rho_{D'}(A) + \delta \\ &\leq e^\varepsilon \rho_{D'}(A) + (e^\varepsilon + 1)\gamma + \delta.\end{aligned} \tag{D.8}$$

$\square$

Proposition 13 is a straightforward application of Corollary 1 of Dalalyan (2014) [4]. For the sake of completeness, we give the statement.

**Theorem 21** ([4], Corollary 1). Assume that $U : \mathbb{R}^d \to \mathbb{R}$ $(d \geq 2)$ is a $m$-strongly convex and $M$-smooth function such that $\int_{\mathbb{R}^d} \exp(-U(\theta))\mathrm{d}\theta < \infty$. Let $\mu \in \mathcal{M}_+^1$ be a Gibbs measure with density $\exp(-U(\theta))/\int \exp(-U(\theta))\mathrm{d}\theta$, and $\gamma \in (0, 1/2)$ be an approximation level. Let the time horizon $S$ and the step-size $h$ be defined by

$$S = \frac{4\log(1/\gamma) + d\log(M/m)}{2m}, \quad h = \frac{\gamma^2(2\alpha - 1)}{M^2 S d\alpha}, \tag{D.9}$$

where $\alpha = (1 + MdS\gamma^{-2})/2$. Then the output of the $T = \lceil S/h \rceil$-step LMC algorithm with initial distribution $N(\theta^*, M^{-1}I_d)$ attains a $\gamma$-approximation of $\mu$ in the total variation distance.

*Proof of Proposition 13.* Set $\alpha = 1$ in Theorem 21. The desired result follows from Proposition 12. $\square$

## Footnotes

[1]The total variation distance between $p_{\theta_1}$ and $p_{\theta_2}$ is given by $|\sigma(\theta_1) - \sigma(\theta_2)|$.

# References for supplementary material

[1] P. Alquier, J. Ridgway, and N. Chopin. On the properties of variational approximations of Gibbs posteriors, 2015. Available at `http://arxiv.org/abs/1506.04091`.

[2] P. Billingsley. *Convergence of Probability Measures*. Wiley, second edition edition, 1999.

[3] O. Catoni. *Pac-Bayesian Supervised Classification: The Thermodynamics of Statistical Learning*. IMS, 2007.

[4] A. Dalalyan. Theoretical guarantees for approximate sampling from smooth and log-concave densities, 2014. Available at `http://arxiv.org/abs/1412.7392`.

[5] R. Hall. *New Statistical Applications for Differential Privacy*. PhD thesis, Carnegie Mellon University, 2013.

[6] B.J.K. Kleijn and A. W. van der Vaart. The Bernstein–von-Mises theorem under misspecification. *Electoric Journal of Statistics*, 6:354–381, 2012.

[7] D. Mir. Differentially-private learning and information theory. In *PAIS*, 2012.

[8] A. W. van der Vaart. *Asymptotic Statistics*. Cambridge University Press, 1998.

[9] Y. Wang, S. Fienberg, and A. Smola. Privacy for free: Posterior sampling and stochastic gradient monte carlo. In *ICML*, 2015.