[Reviews · NeurIPS 2016]

Reviewer 1

Summary

Exponential mechanism is a universal differentially private algorithm, i.e., every differentially private algorithm can be represented as an exponential mechanism. Exponential mechanism requires defining a loss (score) function over the output space parameterized by the data set, and then sampling a candidate from a distribution induced by the score function. Often times the loss function is required to be of bounded range for the privacy analysis. This paper tries to alleviate this concern by sampling from a Gibb’s posterior distribution, with strongly convex prior. The strong convexity of the prior helps to control the tail behavior of the sampling distribution, and hence obtain a (\epsilon,\delta)-differential privacy guarantee.

Qualitative Assessment

One major concern I have with the paper is the amount of complexity introduced. I am not convinced whether a simpler algorithm / analysis would suffice. In the following, I qualify my above statement. In the notation of the paper, let L = \sum \ell(\theta;d_i). If \theta \in R_\Theta (a bounded convex set) and \ell is L-Lipschitz, then one can show that sampling from exp(-\eps/\beta L(\theta)) is \eps differentially private (see [3]). Furthermore, adding a strongly convex regularizer should remove the bound on R_\Theta. by the same argument described in this paper. How does this compare to Section 3.2 in this paper? On a general note, how the utility guarantees of this paper compare to the existing results [8, 14, 3]? More specific comments are as follows: 1. I believe the idea of introducing a strongly convex prior to control the tail distribution of exponential mechanism is an interesting idea. Also the analysis to control the tail behavior of exponential mechanism (in order to obtain a bound on the privacy parameter \delta), seems novel in my opinion. 2. I am concerned with the title mentioning 'without sensitivity'. It seems too catchy to be factually correct. The paper does assume a bound on the gradient of the loss function (instead of the value of the loss function). Even the bound on the gradient can be considered as sensitivity. 3. In Line 31, the paper forgot to mention that [3] also does posterior sampling on the output distribution via exponential mechanism. 4. The exposition of the shrinkage concept (in lines 57-61) can be improved for readability by describing it in terms of regularization. 5. The related work requires significant work. The paper needs to describe the prior work in much more detail in order to place the novelty of their results in perspective. The area of private convex optimization is very well studied. 6. In Line 153, I am concerned with the value of the delta parameter. Typically, for semantic correctness of differential privacy \eps is a small constant (around 0.1) and \delta is a value much small than 1/ n^2 (n is the number of data samples). 7. The bound in Line 199 needs more justification. Since in machine learning typically \lambda is chosen to be around 1/\sqrt n, \beta seems to scale as n^{1/4}. Is this good or bad ? 8. In Line 42 it should be analysis instead of analyzes.

Confidence in this Review

3-Expert (read the paper in detail, know the area, quite certain of my opinion)


Reviewer 2

Summary

The exponential mechanism is a powerful technique in differential privacy that privately selects a parameter theta that approximately minimizes any bounded, "low-sensitivity" loss function. This paper shows that under certain natural conditions, the exponential mechanism remains private when the loss function is unbounded. Specifically, the paper considers special cases of convex empirical risk minimization (where the loss function is the empirical risk).

Qualitative Assessment

This paper considers an interesting question, which is whether the exponential mechanism can be used in settings where the loss function to be minimized is unbounded. The main result is to use the connection between Gibbs sampling and the exponential mechanism to show that when the prior is log-strongly-concave then the exponential mechanism will satisfy differential privacy even when the loss function is unbounded. (The paper distinguishes between eps-DP and (eps,delta)-DP but eps-DP is an artificial special case of differential privacy. In my opinion, a more natural way to present the results is to say that the paper shows that the algorithm is differentially private, and previous works showed that this could not be done for the special case of eps-DP.) I believe the paper cannot be accepted in its current form because there is no analysis of utility. However, I think this is a direction worth pursuing. I recommend rewriting the paper with an analytic or empirical analysis of utility and some comparison to other methods. One challenge is that other methods often assume a bounded loss function, so arguably there are no directly comparable methods. However, one can easily shoehorn unbounded loss into bounded loss for most natural case. For example, the bernoulli loss function is only unbounded if p is close to 0 or 1. You can get a bounded loss function with small loss of utility by requiring that p in (a, 1-a) for some appropriately chosen a. It's not obvious to me whether it's better to use the analysis from this paper or just use truncation and apply results from previous papers. I would like to see the presentation modified to incorporate a utility analysis. The author's rebuttal stating that the utility analysis follows immediately from prior work was very helpful. However, my concern is that the utility will actually depend on the privacy parameters, so without a utility statement, the current submission is not self-contained. That is, if this submission proves that the exponential mechanism is private only when the privacy parameter is "too small," then there may be no utility, but I would have to look up the utility analysis from prior work to determine the meaning of "too small." I now believe that this is a simple issue to correct and I recommend acceptance provided that the author includes a suitable theorem statement in the paper (even if the "proof" just says that the theorem follows immediately by combining bounds from prior work with the new results of this submission).

Confidence in this Review

3-Expert (read the paper in detail, know the area, quite certain of my opinion)


Reviewer 3

Summary

The paper studies privacy guarantees of Gibbs posterior sampling, and obtained PAC-Bayesian-style bounds for unbounded classes, by making assumptions on the prior distribution and loss function. The technical contribution in this paper is two-fold: for non-strongly-convex loss, the authors gives bounds on beta depending on Lipschitz constant; for strongly convex loss, they also derived bounds under additional assumptions on uniform boundedness of posterior mean and local boundedness of loss function gradients. Both of the results are obtained by assuming log-strongly-concave priors.

Qualitative Assessment

It is exciting to see privacy being achieved with unbounded uniform sensitivity measures, by choosing thin-tailed priors for which the posterior KL divergence is well-controlled, which overcomes privacy leakage resulted from extreme value of parameters. The results presented in this paper are interesting and technically sound, and provide some novel insights into the privacy approaches via posterior sampling. Despite the above merits, there are also a few issues in this paper that need improvement: 1. Intuitively Assumption 9 requires only uniform boundedness of posterior expectation and local Lipschitzness of loss function. However, no rigorous evidence was given on the appropriateness of this assumption. The author may need to provide a handful of nontrivial examples with unbounded range and strongly-convex loss, which satisfies this assumption with tight parameters. 2. Getting rid of uniform sensitivity is surely a good news. However, since in practice boundedness can be easily satisfied by confining the norm of parameters and discarding thin tails, it would be important to decide which framework to use. To make their argument more convincing, the authors may also need to compare their results with classical sensitivity-based ones in terms of accuracy. It would be helpful to provide some natural examples that illustrate the superiority of their methods. 3. It would be better if the author can prove something about optimality/tightness of their results, either for bounds or for assumptions. For example, is it necessary for the tail of prior to diminish with sub-Gaussian rate? Is it possible to prove similar results for log-concave or exponential tail distributions? 4. There are a few typos, grammatical errors and formatting issues in this paper.

Confidence in this Review

2-Confident (read it all; understood it all reasonably well)


Reviewer 4

Summary

This paper extends the analysis of the well-known exponential mechanism from the differential privacy literature. This analysis generalizes the analysis of the Gaussian mechanism. In particular, it considers Lipschitz loss functions, rather than bounded loss functions (assuming the loss is a sum over the dataset elements). This requires shifting from pure eps-differential privacy to approximate (eps,delta)-differential privacy.

Qualitative Assessment

Extending the analysis of the exponential mechanism is a very natural question and this paper presents some nice results in this direction. The results are a generalization of existing analyses (such as of the Gaussian mechanism) using less well-known concentration inequalities. The results are pleasing, although not surprising or particularly difficult. The paper is well-written and I believe it would be an excellent addition to the NIPS program. Comments to author: Missing spaces in figure 1 caption and on lines 83, 85, 142, 169, 202. Missing comma on line 208. line 114: theta should be p. Part (i) of Corollary 8 is extremely weak (delta close to 1 ). It might be better to simply remove it.

Confidence in this Review

3-Expert (read the paper in detail, know the area, quite certain of my opinion)


Reviewer 5

Summary

The authors analyze the popular Gibbs sampling algorithm from a privacy-preserving learning perspective. It is shown that when the loss function satisfies certain Lipschitz continuous conditions and the prior distribution is strongly log-concave, the resulting Gibbs sampler satisfies (eps,delta)-differential privacy for free. This result generalizes previous work of Wang et al. that requires the loss function to be uniformly bounded.

Qualitative Assessment

Overall recommendation: poster Though the idea of this paper is rather simple (an unrevised Gibbs sampler and a KL concentration argument), I feel it is worth publishing because it could potentially benefit the practice of private learning. In particular, exponential mechanism is one of the algorithms that could lead to reasonable performance since it avoids explicitly invoking advanced composition within a learning procedure, and extending such a mechanism to adapt to a weaker (and hence more practical) (eps,delta) privacy criterion is certainly relevant. I feel pity that this paper does not have experimental results to empirically verify the practicability of the proposed mechanism. In particular, I would be rather interested to see how the proposed algorithm (using the derived epsilon-delta setting of parameters) could be applied to standard learning problems like logistic regression on a real data set and compare against, for example, the approach given in Want et al. with parameters set at the stronger (eps,0) private regime. Such empirical comparisons would be very useful and shed light on future directions of making differential privacy more practical.

Confidence in this Review

2-Confident (read it all; understood it all reasonably well)


Reviewer 6

Summary

The paper extends the analysis (\varepsilon,\vardelta)-differential privacy when the used Gibbs posterior is generated unbounded loss.

Qualitative Assessment

Differential privacy is an important variant of analyzing sensitivity of estimation results and as such it is worth of a deep inspection. Your results belong to this domain and as such they are valuable. However, the presentation of your results is not good and can hardly attract the appropriate attention of NIPS audience. Bellow, I will give the (incomplete) list of examples, which support this statement and hopefully will help you to create a good future publication. - The basic notions related to differential privacy (which are not definitely common), their meaning and use are poorly commented: Moreover, they appear after your basic Claim. - The connections with robust and Sanov-type analysis developed predominantly in connection with information theory and Bayesian statistics are completely neglected. Your use of Kullback-Leibler divergence strongly indicates usefulness of inspecting them. - The definition of the notion M-smooth function is unfinished (here and on other places it induces impression that you "cut and paste" from your other work; e.g. see the lacking relation of $f(\cdot,D)$ and $f(.,D')$ under (3) to definition (3)) - Mathematical level varies significantly through the paper. For instance, you have used slang-term "parameter" instead of estimator; you assume implicitly independent observations and parsimonious observations; you use vague statements (arbitrary prior) about prior probabilities,; you neglect that almost-sure variants of your statements would be appropriate. At the same time, you enter the quite advanced domains leading to tail-bound inequality (but without sufficient rigor what is the argument $t$ of the function $\alpha(t)$) or logarithmic Sobolev inequality. It is unclear what you expect from the reader. - Insufficient care about mathematical rigor makes some statements too vague and forces the reader to check them in original references. For instance, in Theorem 4, point (b) seems to allow $L$ be $\theta$ dependent, which cannot be the case. - Not always you managed unify notation (e.g. $\alpha$ is used both as a function and constant, $A$ as the measurable set and constant). Some symbols are undefined at all (e.g. $y$ in Figure 1). - The red curve of the gradient in Figure 1 can hardly be gradient of the corresponding to the red loss. - The need for your extension to unbounded losses often disappears by considering prior probability with support in a bounded ball (as you did before Theorem 10). - The dependence of prior probability (12) on $n$ is quite unusual and going against the notion of prior. - The bound (17) seems to be practically useless due to its $\mathcal{O}(n^{2})$ character. - No open problems are mentioned.

Confidence in this Review

2-Confident (read it all; understood it all reasonably well)